# Protective Effects of *Ulva lactuca* Polysaccharide Extract on Oxidative Stress and Kidney Injury Induced by D-Galactose in Mice

**DOI:** 10.3390/md19100539

**Published:** 2021-09-25

**Authors:** Qian Yang, Yanhui Jiang, Shan Fu, Zhaopeng Shen, Wenwen Zong, Zhongning Xia, Zhaoya Zhan, Xiaolu Jiang

**Affiliations:** 1School of Food Science and Engineering, Ocean University of China, No. 5, Yushan Road, Qingdao 266003, China; 15205323877@163.com (Q.Y.); 21190731142@stu.ouc.edu.cn (S.F.); 2Marine Biomedical Research Institute of Qingdao, No. 23, Hong Kong Eastern Road, Qingdao 266071, China; jiangyanhui@ouc.edu.cn (Y.J.); 17864272822@163.com (W.Z.); 3School of Medicine and Pharmacy, Ocean University of China, No. 5, Yushan Road, Qingdao 266003, China; 4Hainan Xin Kaiyuan Pharmaceutical Technology Co., Ltd., Hainan 570311, China; yundiao2009@163.com; 5Fujian Blue Sea Food Technology Co., Ltd., Fujian 355200, China; yundiao2010@163.com; 6State Key Laboratory of Bioactive Seaweed Substances, Qingdao Brightmoon Seaweed Group Co., Ltd., Qingdao 266400, China

**Keywords:** *Ulva lactuca*, polysaccharide, D-galactose, oxidative stress, kidney

## Abstract

Reactive oxygen species (ROS) are the key factors that cause many diseases in the human body. Polysaccharides from seaweed have been shown to have significant antioxidant activity both in vivo and in vitro. The ameliorative effect of *Ulva lactuca* polysaccharide extract (UPE) on renal injury induced by oxidative stress was analyzed. As shown by hematoxylin–eosin staining results, UPE can significantly improve the kidney injury induced by D-galactose (D-gal). Additionally, the protective mechanism of UPE on the kidney was explored. The results showed that UPE could decrease the levels of serum creatinine (Scr), blood urea nitrogen (BUN), serum cystatin C (Cys-C), lipid peroxidation, protein carbonylation, and DNA oxidative damage (8-OHdG) and improve kidney glutathione content. Moreover, UPE significantly increased the activities of superoxide dismutase and glutathione peroxidase and total antioxidant activity in mice. UPE also decreased the levels of inflammatory cytokines TNF-α and IL-6. Further investigation into the expression of apoptotic protein caspase-3 showed that UPE decreased the expression of apoptotic protein caspase-3. These results indicate that UPE has a potential therapeutic effect on renal injury caused by oxidative stress, providing a new theoretical basis for the treatment of oxidative damage diseases in the future.

## 1. Introduction

Reactive oxygen species (ROS) are substances produced by human cells and tissues during normal physiological metabolism. Under normal circumstances, the body’s antioxidant defense mechanisms play a role in the body’s oxidative balance. However, when a large amount of ROS accumulates in the human body, it will cause oxidative imbalance and further aggravate the damage to the body [1]. The accumulation of ROS in the organism comes not only from the damage to the organism itself, but also from environmental pollution, radiation, and abuse of chemical products, which can cause the increase in ROS. Therefore, maintaining a balance of reactive oxygen species and antioxidant systems in the body is crucial to health. D-Galactose (D-gal) is a substance naturally occurring in the human body, but high concentrations of galactose will produce a large number of reactive oxygen species, leading to the aging of multiple tissues in the body and the occurrence of age-related immune decline; degenerative nervous system disorders; and damage to pancreas, kidney and other organs [2]. Therefore, the D-gal mouse model has become a widely used aging model.

Natural products with antioxidant activity have received extensive attention due to their good biological safety. With the attention to marine resources and the development of marine natural products, a variety of products that are raw materials with seaweed polysaccharides are gradually entering people’s line of sight, and the antioxidant effect of algae polysaccharides has been confirmed [3]. It has been shown that fucoidans from brown algae can not only reduce the production of reactive oxygen species, but also increase the level of glutathione (GSH) and catalase (CAT) activity and reduce cytotoxicity [4,5]. *Ulva lactuca* is a kind of green algae in coastal areas of China. As a traditional Chinese medicine, it has good healthcare function. Ulvan is a water-soluble polysaccharide existing in the cell wall of *Ulva*, accounting for 8–29% of the dry weight [6]. It is a kind of sulfated heteropolysaccharide, which mainly includes sulfated rhamnose (Rha), xylose (Xyl), and uronic acid (glucuronic acid (GlcA) and iduronic acid (IdoA)); the structures of the main disaccharide repeating unit are [β-D-GlcpA-(1 → 4)-α-L-Rhap3s] and [α-L-IdopA-(1 → 4)-α-L-Rhap3s] [7]. Li et al. [8] found that the ulvan had a good scavenging ability against superoxide free radicals and DPPH free radicals in a concentration-dependent manner. Marlene Godardd et al. [9] reported that *U. lactuca* polysaccharide could significantly increase the activities of superoxide dismutase (SOD) and glutathione peroxidase (GSH-Px) and inhibit the production of lipid peroxidation and superoxide anion. In addition, ulvan with high sulfate content has obvious antioxidant ability and lipid-lowering activity [10]. Therefore, using ulvan as the raw material to develop medicine or food that can remove excess free radicals in the body is not only safe in composition but also effective and has a broad prospect.

In this study, *Ulva lactuca* polysaccharide extract (UPE) was obtained from *U. lactuca*, and D-gal was used to induce oxidative stress in mice to study the effects of UPE on oxidative stress and inflammation in mice. Hematoxylin and eosin staining and immunohistochemistry were used to explore the tissue damage and cell apoptosis in mice.

## 2. Results

### 2.1. Characterization of UPE

As shown in Table 1, the main component of UPE is carbohydrate, accounting for about 52.61% of the total composition; in addition, UPE contains uronic acid and sulfate group. By molecular weight analysis, the average molecular weight of UPE is about 891.25 kDa. According to the analysis of monosaccharide composition (Figure 1b,c), it is mainly composed of Rha, GlcA, glucose (Glc), and Xyl, and the content of Rha is the highest (45.33%). In addition, through FT-IR (Figure 1a), the signals at 1640, 1427, and 1054 cm^−1^ are respectively the absorption peaks of the stretching vibration of carbonyl group C=O, the absorption peaks of the stretching vibration of carboxyl group C-O, and the absorption peaks of the O-H angular vibration. These three absorption peaks are characteristic absorption peaks of uronic acids [11]. The absorption peaks of 1259 and 843 cm^−1^ are the characteristic absorption peaks of the sulfate group, where 1259 cm^−1^ is the absorption peak of S=O stretching vibration and 843 cm-1 is the absorption peak of C-O-S stretching vibration. According to the above results, it was proved that both sulfuric acid and uronic acid were present in UPE, which were the main characteristics of ulvan.

### 2.2. Effect of UPE on Organ Index

Organ index is the ratio of a certain organ to its body weight. After an animal’s body is damaged, organ weight will change, and organ coefficient will also change. Therefore, the viscera ratio can intuitively judge the severity of body damage. The body weight and kidney weight of mice are shown in Table 2. The renal index of mice is shown in Figure 2a. The renal index of mice injected with D-gal decreased significantly (*p* < 0.01) when the initial body weight was basically the same. However, after UPE treatment, the renal index was significantly increased (*p* < 0.01), which indicated that UPE alleviated the phenomenon of renal atrophy to a certain extent and had a protective effect on the kidney.

### 2.3. Effects of UPE on Serum Indexes

The levels of Scr, BUN, and Cys-C in serum and kidney are shown in Figure 2b–d. Compared with the control mice, the contents of Scr, BUN, and Cys-C in serum of mice treated with D-gal were increased by 42.06%, 19.94%, and 25.43% (*p* < 0.01), respectively. When feeding low-dose UPE, the contents of Scr and Cys-C were significantly decreased (both *p* < 0.01) compared with the NC group, but there was no statistical significance in the change of BUN level. However, the contents of Scr, BUN, and Cys-C in the high-dose group were significantly decreased (28.76%, 12.65%, and 16.76%, respectively, *p* < 0.01), indicating that UPC had a protective effect on kidney injury induced by D-gal in mice.

### 2.4. Effects of UPE on Renal Oxidation and Antioxidant Indexes

The large amount of ROS produced by oxidative stress will cause oxidative damage to biological macromolecules such as proteins, lipids, and DNA and then cause damage to the body. The oxidant and antioxidant indexes of mice are shown in Figure 3. The contents of MDA, protein carbonyl, and 8-OHdG in mice in the D-gal group were significantly increased (52.46%, 31.48%, and 27.45%, respectively) compared with those in the BC group (all *p* < 0.01), and the content of GSH decreased significantly (23.16%, *p* < 0.05). Moreover, D-gal also led to a decrease in the activities of SOD (16.66%), GSH-Px (21.74%), and T-AOC (47.99%) in mice (all *p* < 0.01). After using low-dose UPE, the contents of MDA, protein carbonyl, and 8-OHdG in mice were reduced by 10.79%, 13.78%, and 9.72%, respectively (*p* < 0.01), and the activities of SOD, GSH-Px, and T-AOC were also significantly increased (10.51%, 46.65%, and 11.97%, respectively), but the change in GSH content was not statistically significant (*p* > 0.05). In contrast, in the high-dose group, MDA, protein carbonyl, and 8-OHdG were also significantly reduced (all *p* < 0.01); GSH content was significantly increased (43.18%, *p* < 0.05); and the activities of SOD, GSH-Px, and T-AOC were close to the level of normal mice. These results show that UPE can significantly reduce oxidative damage caused by D-gal and improve the antioxidant capacity of mice.

### 2.5. Effects of UPE on Inflammatory Factors

The levels of inflammatory cytokines in mice are shown in Figure 4. Compared with the BC group, the levels of IL-6 and TNF-α in mice injected with D-gal were significantly increased (*p* < 0.01). The levels of IL-6 and TNF-α in mice in the low-dose group were slightly decreased, but there was no statistically significant difference (*p* > 0.05). The IL-6 level in the high-dose group showed no significant change, while the TNF-α level significantly decreased.

### 2.6. Histopathological Analysis

The most direct method to judge the pathological injury of the kidney is to observe the renal tissue morphology of mice by microscope. As shown in Figure 5, the glomeruli of mice in the normal group were round, with obvious renal cysts, and renal tubules were arranged neatly with obvious lumens, without pathological changes. However, D-gal-treated mice had irregular glomerulus shape, with some degree of atrophy, irregular renal capsule lumen, and fuzzy renal tubules. However, after UPE treatment, the glomeruli and renal lumen were normal, the margins of the lumen were clear, and the renal tubules and lumen were distinguishable. The histomorphological characteristics of renal sections stained with H&E showed that UPE had a protective effect on D-gal-induced renal injury.

### 2.7. Expression of Caspase-3 Protein in Kidney

During apoptosis, the caspase-3 protein was positively expressed, and the immunoreactive product was stained in renal tubules. The expression of the caspase-3 protein in the mouse kidney is shown in Figure 6. Apoptosis was observed in each group. Compared with the blank group, caspase-3 showed obvious positive expression in the NC group, and the positive cells were increasingly dense. The positive expression of UPE was significantly reduced in mice after UPE treatment, suggesting that UPE may have a protective effect on apoptosis. Image J software was used to analyze the staining situation, and it was found that compared with the BC group, the protein expression level of the NC group was significantly increased (*p* < 0.05), and the expression level of caspase-3 decreased significantly after treatment (*p* < 0.05), which further proved the improvement effect of UPE on apoptosis.

## 3. Discussion

The kidney is an important organ in the human body. Through filtration, the kidney educts the excess waste in the human body through urine, so as to maintain the metabolism of the human body and ensure the stability of the internal environment of the body. It has been reported that the production of large amounts of reactive oxygen species is the key pathogenic factor for a variety of kidney diseases [12]. In the event of oxidative stress, both renal tubules and vascular cells produce large amounts of reactive oxygen species, which attack renal cells and tissues and further cause renal damage [13]. In this study, in order to investigate the antioxidant ability of ulvan in vivo, we established a model of oxidative stress induced by D-gal to study the protective effect of UPE on kidney injury induced by oxidative stress.

Previous studies have shown that D-gal can lead to decreased indexes of kidney, thymus, brain, and other organs [14,15]. The results showed that UPE significantly improved the kidney atrophy induced by D-gal, which proved the protective effect of UPE on kidneys to a certain extent. This result was also confirmed in the H&E observation. D-gal-induced oxidative stress can cause damage to various tissues and organs. In mice treated with D-gal, capillary congestion of glomeruli and renal tubules would occur, and renal tubules would degenerate and die, leading to renal injury [16]. In our study, changes in renal structure were observed in mice. Compared with D-gal-treated mouse kidneys, UPE improved the pathological phenomena of glomerular atrophy and renal lumen obscuration, suggesting that UPE had a certain therapeutic effect on mouse kidney injury induced by D-gal treatment.

Excessive production of reactive oxygen species can cause oxidative damage to biological macromolecules such as proteins, nucleic acids, lipids, and DNA [17]. Lipid is the most common target of oxidative stress, and its oxidation product MDA can cause serious damage to mitochondrial respiratory chain complex and cell membrane, and it is widely used as a marker of lipid peroxidation [18]. Although 8-OHDG is not a specific marker of oxidative damage, ROS and hydroxyl radicals can attack DNA to produce 8-OHDG, which is the most commonly used biomarker of oxidative damage of DNA [19]. The carbonyl group is a key marker of protein oxidation, which is formed because reactive oxygen species directly attack the free amino group in amino acid molecules [20]. GSH is an important antioxidant and free radical scavenger in the body, which can react with H_2_O_2_ under the catalysis of GSH-Px to form GSSH, and remove peroxides and hydroxyl radicals produced by cellular respiration metabolism [15]. As an important component of the body’s antioxidant defense system, SOD is the primary material for scavenging free radicals in the body, which can convert superoxide free radicals into hydrogen peroxide. GSH-Px protects cells from damage by scavenging lipid and hydrogen peroxides [21]. T-AOC is an evaluation index of comprehensive antioxidant capacity. The changes of oxidation products (MDA, 8-OHdG, protein carbonyl group, and GSH) and antioxidant enzyme activities (SOD, GSH-Px, and T-AOC) can be used to determine the oxidation level in the body. Through our study, it was found that the levels of MDA, 8-OHdG, and protein carbonyl in mice after UPE treatment were significantly reduced, suggesting that UPE has a significant therapeutic effect on oxidative injury. At the same time, we also found that UPE could significantly increase the activities of SOD, GSH-Px, and T-AOC in mice, which was consistent with the study of Liu et al., which has shown that polysaccharide extracted from *U. lactuca* could improve the antioxidant capacity of SAMP8 mice and also reduce the contents of inflammatory factors such as TNF-α, IL-6, and IFN-γ in mice [22]. These results provide evidence that UPE can be used to treat oxidative stress.

Kidney damage is followed by problems with kidney function. Scr, BUN, and Cys-C are commonly used to measure kidney health, and their concentrations are increased by kidney injury or renal failure [23]. The creatinine and urea in the blood are filtered mainly in the glomerulus, part of which is filtered out of the blood and the rest of which is reabsorbed. Serum Cys-C has a better effect in marking glomerular filtration level and can therefore be used as a marker of glomerular filtration rate [24]. In this study, after acute kidney injury, Scr, BUN, and Cys-C levels were significantly increased, indicating a sharp decrease in renal tubular filtration rate, which was consistent with previous studies [25]. Based on these conclusions, the contents of serum creatinine, blood urea nitrogen, and Cys-C decreased significantly after treatment with UPE and approached the normal level with the increase in the concentration, indicating that the filtration ability of mouse kidneys gradually recovered. Studies have shown that alginate oligosaccharides have an obvious protective effect on kidney injury induced by D-gal in mice, and the levels of BUN and Scr are improved after 4 weeks of treatment [26]. Kelp polysaccharide can reduce the content of Scr and BUN. This protective effect may be related to the anti-inflammatory and antioxidant effects of sulfated polysaccharides [27]. Therefore, it was speculated that the protective effect of UPE on the kidney might be related to its antioxidant capacity.

Oxidative stress can activate a variety of transcription factors, such as NF-κB, AP-1, and P53, which lead to the expression of inflammatory cytokines, anti-inflammatory molecules, and other genes [28]. TNF-α is a major inflammatory cytokine that can kill target cells, promote cell apoptosis, and participate in local inflammation and activation of endothelial cells. IL-6 is a type of interleukin that acts as a proinflammatory cytokine and an anti-inflammatory myosin. In our study, the levels of IL-6 and TNF-α in mice in the BC group were significantly higher than those in the control group, indicating a severe inflammatory response in mice. It may be caused by the production of proinflammatory mediators stimulated by oxidative stress. Although the content of IL-6 in mice was decreased after using UPE, the effect was not obvious, but the level of TNF-α in mice was decreased significantly (*p* < 0.05). Some studies have shown that *U. lactuca* polysaccharide can significantly reduce the serum levels of TNF-α and NO in breast cancer mice, indicating that UPE can regulate inflammatory response and reduce the level of inflammatory cytokines [29]. It has also been reported that *Gracilaria* extract achieves anti-inflammatory effects by reducing the levels of NO, IL-6, and TNF-α [30]. Purified kelp polysaccharide can reduce inflammatory reactions by inhibiting the activation of the TGF-β1-mediated inflammatory cytokine signaling pathway, downregulating the levels of TNF-α and IL-1β, etc. [31]. It was suggested that UPE might regulate the expression of cytokines at the transcriptional level [32].

Apoptosis is closely related to oxidative stress induced by ROS. Oxidative stress can mediate apoptosis by stimulating the synthesis of inflammatory factor TNF-α. TNF-α can activate caspase-8 after binding to surface receptors and then shear and activate caspase-3, leading to the production of apoptosis [33]. In addition, excessive ROS will directly attack the mitochondrial membrane, change the ratio of proapoptotic protein Bax and anti-apoptotic protein Bcl-2, mediate the release of cytochrome C, activate apoptotic factor caspase-3, and cause cell apoptosis [34]. Cascade activation of the cysteine protease family is an essential procedure in apoptosis, in which caspase-3 plays a key role [35]. In this study, the renal tubular epithelial cells of mice treated with D-gal showed a strong positive reaction, the expression of caspase-3 was enhanced, and abnormal renal apoptosis occurred. However, after UPE treatment, the apoptosis of renal cells was significantly reduced, which indicated that UPE had a certain protective effect on the kidney. It has been reported that *Enteromorpha prolifera* polysaccharides can regulate the mitochondrial apoptosis pathway and reduce cell damage [36]. Astragalus polysaccharides can also protect acute kidney injury by regulating oxidative stress and improving mitochondrial apoptosis signals [37]. These results indicate that UPE has a good effect on apoptosis induced by D-gal and provide a direction for the development of UPE in the future.

## 4. Materials and Methods

### 4.1. Chemicals

D-gal and ascorbic acid (VC) were supplied by Sinopharm Chemical Reagent Co., Ltd. (Guangzhou, China). Assay kits for the measurements of superoxide dismutase (SOD), glutathione peroxidase (GSH-Px), glutathione (GSH), malondialdehyde (MDA), total antioxidant capability (T-AOC), serum creatinine (Scr), blood urea nitrogen (BUN), serum cystatin C (Cys-c), 8-hydroxylated deoxyguanosine (8-OHdG), and protein carbonyl were purchased from Jiancheng Bioengineering Institute (Nanjing, China). ELISA detection kits for IL-6 and TNF-α were obtained from Dakewe Biotech Corporation (Beijing, China).

### 4.2. Sample Preparation

*U. lactuca* was soaked in 80% alcohol for 18 h and heated at 70 °C for 4 h to remove pigment, protein, and some salt. The sample was centrifuged and dried. The polysaccharide was extracted by hot water extraction. The crude extract and water were dissolved in a ratio of 1:30 and reacted at 100 °C for 1 h. Ulvan crude extracts were filtered and centrifuged (4000 rpm, 10 min). The supernatants were collected and concentrated at reduced pressure. They were precipitated overnight at 4 °C with twice the volume of 95% ethanol. The polysaccharide was obtained by freeze-drying after collecting the precipitation and named UPE.

### 4.3. Characterization of UPE

The content of total sugar in UPE was determined by phenol–sulfuric acid method, and glucose was taken as the standard [38]. Protein content was determined by Kjeldahl method [39]. The uronic acid content in UPE was calculated by m-hydroxybiphenyl method [40]. The sulfate radical content was evaluated by barium sulfate turbidimetry [41].

The molecular weight of UPE was analyzed by gel permeation chromatography (GPC). Agilent 1260 HPLC was used and equipped with a PL aquagel-OH 60 column (300 mm × 7.5 mm; Tosoh, Shiba, Tokyo, Japan) and refractive index detectors. The dried polysaccharide samples were ground and mixed with potassium bromide to make the tablets and scanned by Nexus 70 infrared spectrometer. The monosaccharide components of UPE were determined by reversed-phase HPLC method. After acidolysis with trifluoroacetic acid, PMP was used for derivation. Monosaccharide analysis was performed on a ZORBAX Eclipse XDB-C18 separation column (4.6 mm × 250 mm) at 245 nm.

### 4.4. Animals and Diet

Forty-five eight-week-old male Kunming mice were raised in an experimental environment and subjected to a light–dark cycle at 23 ± 1 °C for 12 h. During the week of domestication, the animals were given normal lab feed and free water. Mice were randomly divided into blank control group (BC), negative control group (NC), positive control group (PC) (VC, 100 mg/kg), low-dose group (LD) (UPE, dose of 50 mg/kg), and high-dose group (HD) (UPE, dose of 300 mg/kg), with 9 mice in each group. Mice in NC group, PC group, LD group, and HD group were subcutaneously injected with D-Gal at a dose of 400 mg/kg, and mice in BC group were injected with 0.9% normal saline for 10 weeks. Starting from week 7, the LD group and HD group were intragastrically given UPE at the corresponding dose, while the PC group was intragastrically given VC once a day for 4 consecutive weeks. The BC and NC groups were given the same volume of distilled water. The experiment design was approved by the Animal Care and Use Committee of Ocean University of China (certificate no. SYXK2012014).

### 4.5. Body Weight Measurement

The mice were weighed every 2 days. On the last day of the experimental period, after 8–12 h of fasting, the mice were sacrificed under ether narcotization. The kidney was isolated and weighed to calculate the organ coefficient, using the following formula:Coefficient (mg/g) = organ weight (mg)/body weight (g).

### 4.6. Serum Indexes Analysis

Blood samples were collected from the eye socket and centrifuged at 4 °C and 3000 rpm for 30 min, and the serum was stored at −80 °C. The contents of Scr, BUN, and Cys-c were determined.

### 4.7. Biochemical Analysis

Mice were sacrificed by cervical vertebrae removal. The kidneys were carefully removed, washed with 0.9% NaCl, frozen in liquid nitrogen, and stored at −80 °C for subsequent physical and chemical analysis. Kidney specimens were homogenized with tissue homogenizer in phosphate buffer pH 7.4, and the supernatant was collected centrifugally (3000 rpm, 10 min). The contents of MDA, SOD, GSH-Px, and T-AOC were measured according to the kit instructions, and the renal antioxidant enzyme activity was evaluated. The contents of protein carbonyl and 8-OHdG were determined to evaluate the oxidative damage level of protein and DNA. The levels of TNF-α and IL-6 in the kidney were determined by ELISA, and the levels of inflammatory factors were evaluated.

### 4.8. Hematoxylin and Eosin (H&E) Staining

Kidney specimens were fixed in 10% formaldehyde normal saline for 5–7 days, washed with clean water for 10 min, and dehydrated with alcohol classification (75%, 85%, 95%, 100%) in accordance with conventional sequence, and then they were treated with xylene to make them transparent. The specimen was then embedded in paraffin. The specimens were cut into 5 mm slices by a slicer. The tissues were stained with hematoxylin and observed under an optical microscope.

### 4.9. Immunohistochemical Analysis

The kidney slices were dewaxed and hydrated by SP immunohistochemical method, the antigens were repaired with citric acid buffer solution, catalase and nonspecific reaction sites were sealed with H_2_O_2_ and goat serum, and the slices were rinsed with PBS. According to the requirements of the kit, the expression of caspase-3 protein in renal tissue was detected. The images were observed with an optical microscope and analyzed with ImageJ software.

### 4.10. Statistical Analysis

Data are expressed as mean ± standard deviation. Statistical analysis was conducted through one-way ANOVA using SPSS 20.0 software (IBM, New York, NY, USA). All data are presented as the mean ± SD, and values of *p* < 0.05 and *p* < 0.01 were considered statistically significant.

## 5. Conclusions

In this study, we demonstrated the antioxidant activity and corresponding inflammatory response of ulvan against oxidative stress induced by D-gal in mice. The results showed that ulvan could significantly decrease the contents of Scr, BUN, and Cys-C in the kidney; increase the glomerular filtration rate; improve the activities of SOD and GSH-Px and total antioxidant capacity in mice; and reduce the damage to biomacromolecules caused by oxidative damage. In addition, it also significantly improved the changes in TNF-α and IL-6 caused by oxidative stress. In the aspect of apoptosis, ulvan could protect against the D-gal-induced apoptosis of renal tubule cells. These results confirm that ulvan has a protective effect on kidney injury caused by oxidative stress. It provides a basis for the development of antioxidants and the treatment of related diseases in the future.

## Figures and Tables

**Figure 1 marinedrugs-19-00539-f001:**
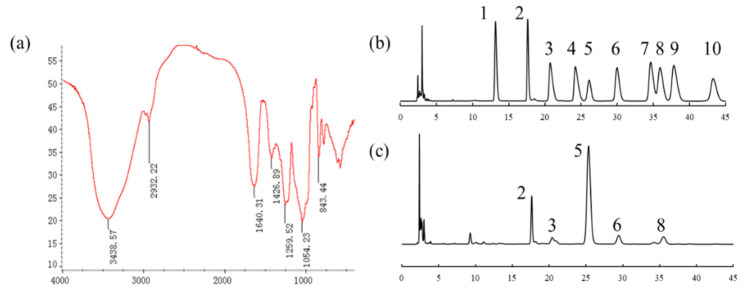
Structural characterization of UPE. (**a**) FT-IR. (**b**) Standard monosaccharide. 1-Man, 2-Rha, 3-GlcA, 4-GalA, 5-Lac (internal standard), 6-Glc, 7-Gal, 8-Xyl, 9-Ara, 10-Fuc. (**c**) Monosaccharide composition of UPE. 2-Rha, 3-GlcA, 5-Lac (internal standard), 6-Glc, 8-Xyl.

**Figure 2 marinedrugs-19-00539-f002:**
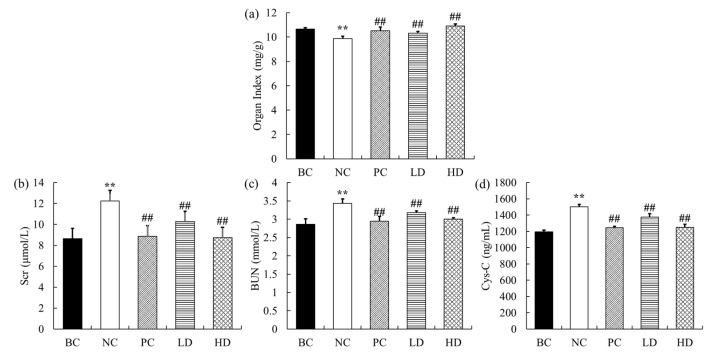
Effect of UPE on organ index and serum Scr, BUN, and Cys-C levels. (**a**) Organ index. (**b**) Scr content. (**c**) BUN content. (**d**) Cys-C content. The data are expressed as mean ± standard deviation (*n* = 9). ** *p* < 0.01 compared with BC group; ## *p* < 0.01 compared with NC group. BC: blank control, NC: negative control, PC: positive control, LD: low-dose UPE (50 mg/kg), HD: high-dose UPE (300 mg/kg).

**Figure 3 marinedrugs-19-00539-f003:**
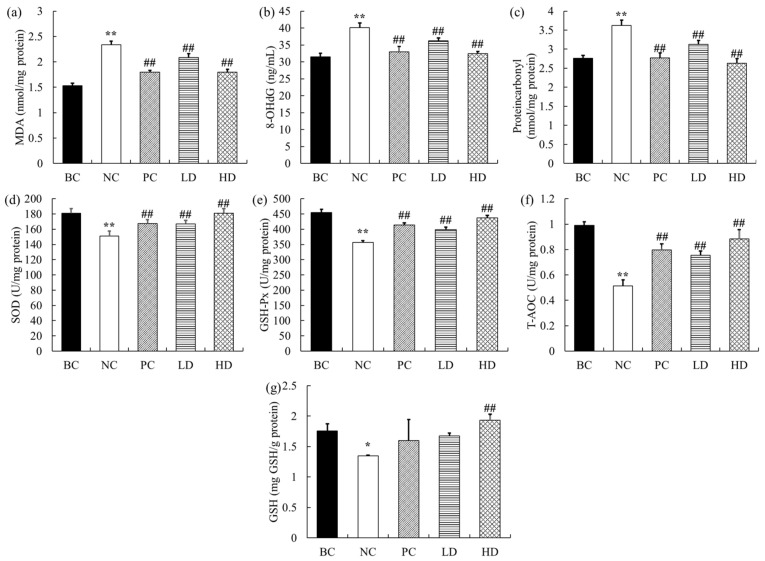
Levels of antioxidant indexes in the kidney of mice. (**a**) MDA content. (**b**) 8-OHdG concentration. (**c**) Protein carbonyl content. (**d**) SOD activity. (**e**) GSH-Px activity. (**f**) T-AOC activity. (**g**) GSH content. Data are presented as mean ± standard deviation (*n* = 9). * *p* < 0.05, ** *p* < 0.01 compared with BC group; ## *p* < 0.01 compared with NC group. BC: blank control, NC: negative control, PC: positive control, LD: low-dose UPE (50 mg/kg), HD: high-dose UPE (300 mg/kg).

**Figure 4 marinedrugs-19-00539-f004:**
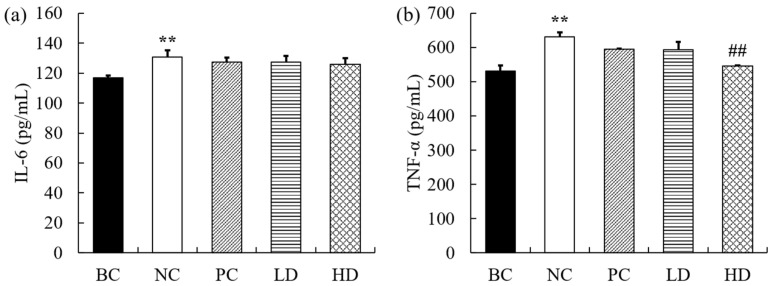
The levels of inflammatory factors in the kidneys of mice. (**a**) IL-6. (**b**) TNF-α. The data are expressed as mean ± standard deviation (*n* = 9). ** *p* < 0.01 compared with BC group; ## *p* < 0.01 compared with NC group. BC: blank control, NC: negative control, PC: positive control, LD: low-dose UPE (50 mg/kg), HD: high-dose UPE (300 mg/kg).

**Figure 5 marinedrugs-19-00539-f005:**
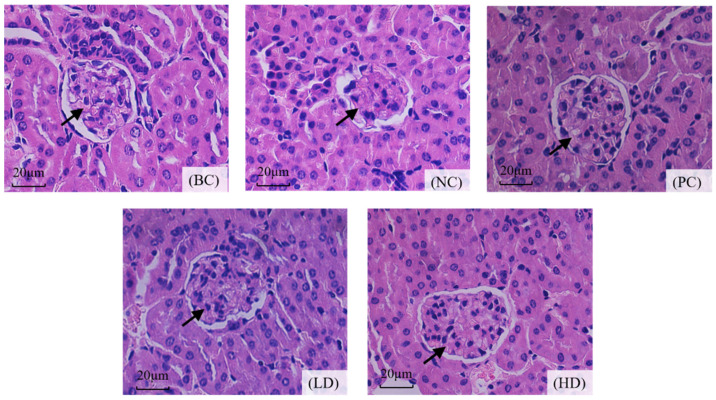
Effect of UPE on histopathological changes of the D-gal-treated kidney. The glomerulus is marked by the arrow. In the NC group, glomeruli are atrophic and deformed, and the renal sac is irregular. The LD and HD groups improved significantly (×400). BC: blank control, NC: negative control, PC: positive control, LD: low-dose UPE (50 mg/kg), HD: high-dose UPE (300 mg/kg).

**Figure 6 marinedrugs-19-00539-f006:**
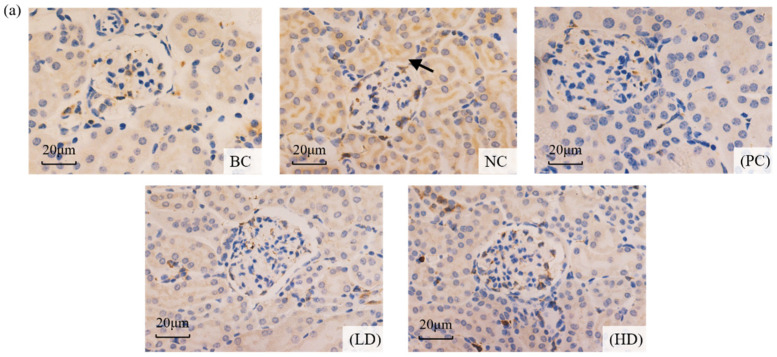
Effect of UPE on the expression of caspase-3 in kidney treated with D-gal. (**a**) The arrow in the image shows positive expression of caspase-3 protein and apoptosis (×400). (**b**) The expression of caspase-3. Data are presented as mean ± standard deviation (*n* = 9). * *p* < 0.05 compared with BC group; # *p* < 0.05 compared with NC group. BC: blank control, NC: negative control, PC: positive control, LD: low-dose UPE (50 mg/kg), HD: high-dose UPE (300 mg/kg).

**Table 1 marinedrugs-19-00539-t001:** Molecular weight, composition, and monosaccharide composition of UPE.

Molecular Weight (kDa)	Monosaccharide Composition (%)		Composition (%)	
891.25	Rha	45.33%	Total sugar	61.98
	GlcA	15.50%	Protein	2.39
	Glc	18.97%	Uronic acid	9.13
	Xyl	20.29%	Sulfate group	16.50

**Table 2 marinedrugs-19-00539-t002:** The different groups for body weights and kidney weights.

	BC	NC	PC	LD	HD
Body Weight (g)	28.125 ± 2.064	27.550 ± 2.263 *	29.300 ± 1.747 ##	29.171 ± 4.085 ##	27.85 ± 2.468
Kidney Weight (g)	0.306 ± 0.023	0.274 ± 0.019 **	0.307 ± 0.022 ##	0.307 ± 0.045 ##	0.304 ± 0.023 ##

* *p* < 0.05, ** *p* < 0.01 compared with BC group; ## *p* < 0.01 compared with NC group. BC: blank control, NC: negative control, PC: positive control, LD: low-dose UPE (50 mg/kg), HD: high-dose UPE (300 mg/kg).

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
