# Peer review of "Protective Effects of Ulva lactuca Polysaccharide Extract on Oxidative Stress and Kidney Injury Induced by D-Galactose in Mice"

_marinedrugs, 2021, doi:10.3390/md19100539_

Round 1

Reviewer 1 Report

The manuscript "Protective effects of Ulva lactuca polysaccharide extract on oxidative stress and kidney injury induced by D-galactose in mice." by Yang et al has been submitted as an Article.

The study analyzed the possible ameliorative effect of ulva polysaccharide extract on renal injury induced by oxidative stress in D-gal mice.

In general, the experiments of the study are sound and the results are convincing.

Please address the following two points:

  • Figure 6: The legend to Figure 6 mentions arrows, which are not clearly visible in the pictures.
  • Figure 6: The expression of the caspase-3 protein is difficult to compare for the different samples. Please add a statistical analysis of the immunohistochemical data or a Western Blot.

Reviewer 2 Report

In this manuscript by Yang et al., the antioxidant activity and the inflammatory response of Ulva lactuca, a natural product, was studied against oxidative stress induced by D-gal in mice. The authors performed various biochemical and immunohistochemical assays and showed that the algae could provide protective effect on kidney injury caused by oxidative stress. In general, the provided information is satisfactory. However, I have some few comments and suggestions:

  1. The authors can reorganize the abstract highlighting the impact of the study giving also a perspective of their extract
  2. Line 71: to induce oxidative stress models in mice to study the effects.. inflammation in mice. Which models? The authors should clarify
  3. Line 97: what is Viscera ratio? The author should explain the meaning.
  4. Which is the exact body weight of animals before and after treatment. Please give a Table or compile 4.5 Section
  5. Figure 2. Please give details about x-axes and differences between bars. Please also repeat for the other Figures.
  6. Line 119. Oxidative stress will cause damage.. This is not totally correct. Please rephrase here.
  7. Line 207: 8-OHdG is not a specific biomarker for oxidative damage due to various artifacts and multiple ROS pathways. The authors could add: Although 8-OH-dG is often formed at higher levels compared to the other lesions, however this lesion is generated by various ROS, including hydroxyl radical. See Chatgilialoglu et al., 2021, doi 1080/10715762.2021.1876855
